# A Neurosymbolic Approach to Natural Language Formalization and Verification

## Abstract

Large Language Models perform well at natural language interpretation and reasoning, but their inherent stochasticity limits their adoption in regulated industries like finance and healthcare that operate under strict policies. To address this limitation, we present a two-stage neurosymbolic framework that (1) uses LLMs with optional human guidance to formalize natural language policies, allowing fine-grained control of the formalization process, and (2) uses inference-time autoformalization to validate logical correctness of natural language statements against those policies. When correctness is paramount, we perform multiple redundant formalization steps at inference time, cross checking the formalizations for semantic equivalence. Our benchmarks demonstrate that our approach exceeds 99% soundness, indicating a near-zero false positive rate in identifying logical validity. Our approach produces auditable logical artifacts that substantiate the verification outcomes and can be used to improve the original text.

## 1 Introduction

The content generation and reasoning capabilities of Large Language Models (LLMs) continue to advance rapidly, demonstrating unprecedented improvements in coherence and analytical accuracy (Wei et al., 2022; Yao et al., 2023; Lewis et al., 2021). Despite these advances, their probabilistic nature and tendency to generate plausible but incorrect information (hallucinations, cf. Xu et al. 2024b) remain barriers to widespread adoption in regulated sectors. Industries such as healthcare, financial services, and legal practices have legal and regulatory obligations for accuracy and auditability that current LLM technology has yet to meet (Haltaufderheide & Ranisch, 2024).

Companies develop institutional policies to ensure compliance with applicable laws and regulations. Such policies are typically captured in natural language (NL) documents that define rules, procedures, or guidelines. A challenge thus emerges when organizations look to deploy LLMs to answer questions about such documents: can we develop *guardrails* to ensure that LLM outputs conform to institutional policies? Consider an airline implementing a chatbot to assist customer service representatives in navigating refund policies: if the chatbot incorrectly claims that a customer is eligible for a refund when they are not, this could lead to legal exposure and loss of customer trust.

An effective guardrail would help representatives decide if they can rely on a chatbot response without spending additional human effort to verify it. The key concern would be to ensure that when the guardrail reports an answer is valid, it actually is. Inspired by the concept of soundness in logic, we define *soundness* as $(1 - p)$, where $p$ is the overall probability of incorrect validity claims. High soundness thus means that across all requests, incorrect approvals are rare. Following established practices in safety-critical systems, where reliability is often measured in "nines" (e.g., 99% = "two nines," 99.9% = "three nines"), we target soundness levels of at least 99%, and secondarily focus on recall to maximize the probability of accepting valid content. While aiming to maximize recall under this requirement, we also pursue actionable feedback that steers LLMs toward content that a conservative guardrail can accept.

A natural candidate for developing robust compliance guardrails are symbolic reasoning systems, as they leverage formal logic to generate independently verifiable guarantees (Robinson & Voronkov, 2001). This approach aligns well with policy documents, which rely on logical, rule-like statements (e.g., "if a flight is canceled or . . . , then passengers are entitled to a refund"). However, symbolic

methods are inherently limited in interpreting natural language, which has triggered the development of neurosymbolic approaches: hybrid solutions that combine the NL processing capabilities of neural networks with the mathematical rigor of symbolic systems (Hitzler & Sarker, 2021).

This paper presents LOGICAL REASONING GUARDRAILS (LRG), a neurosymbolic approach that exceeds 99% soundness on datasets that it was not trained on—an assurance threshold unattainable by existing pure neural or neurosymbolic approaches. This high soundness is also reflected in conventional metrics such as false positive rate and precision, where LRG outperforms competing approaches. In addition, LRG delivers explainable verdicts and provides actionable feedback that LLMs can utilize to refine their outputs. The mechanisms enabling 99% soundness establish a robust foundation for future research aimed at pushing assurance boundaries to three nines and beyond.

LRG operates through two complementary components. The first, called POLICY MODEL CREATOR (PMC), combines LLMs with symbolic reasoning to translate NL policies into formal *policy models* expressed in logic. The process begins with an autoformalization phase that generates an initial policy model. This is followed by an optional vetting phase where domain experts review and refine the policy model with assistance from the system. Through this vetting process, domain experts can resolve ambiguities and inconsistencies that may exist in the original documents, or correct potential omissions and imprecisions from autoformalization. Policy model creation occurs offline, where its computation cost will be amortized across subsequent verification tasks. Notably, policy models serve as enduring sources of truth that provide definitive, unambiguous references within their respective domains.

The second component, called ANSWER VERIFIER (AV), implements a guardrail that verifies NL content against policy models. The AV uses LLMs to translate NL content into individual logical claims over the vocabulary of the policy model. Each claim is analyzed separately and assigned a verification result, together with detailed logical explanations and corrective guidance where applicable. To increase reliability, the AV uses multiple LLMs to simultaneously formalize the same NL content, then uses symbolic reasoning to compare formalizations and assign confidence scores. The AV delivers auditable logical artifacts that substantiate the verification outcomes.

## 2 RELATED WORK

Recent approaches use LLMs as judges to evaluate factual accuracy (Jacovi et al., 2025), though these rely on the same probabilistic models that introduce errors. MiniCheck (Tang et al., 2024) provides efficient fact-checking by decomposing claims. RefChecker (Hu et al., 2024) introduces knowledge-centric verification against structured knowledge bases. SelfCheckGPT (Manakul et al., 2023) leverages consistency across multiple responses to detect hallucinations. FactCheck-GPT (Wang et al., 2024) provides comprehensive evaluation with fine-grained error categorization. While promising, these methods operate within the probabilistic paradigm and cannot provide formal guarantees. Our neurosymbolic framework formally verifies logical validity against explicit policies, achieving near-zero false positives.

Neurosymbolic systems combine LLMs with symbolic reasoning, typically translating natural language to formal representations solved by external reasoners (Pan et al., 2023; Olausson et al., 2023; Callewaert et al., 2025; Ryu et al., 2024). Some leverage LLMs' reasoning through Chain-of-Thought (Wei et al., 2022) or hybrid approaches (Xu et al., 2024a; Liu et al., 2025; Xiong et al., 2024). Notable systems include approaches based on Answer Set Programming (ASP) (Ishay et al., 2023; Yang et al., 2023; Brewka et al., 2011) that generate ASP representations. LINC (Olausson et al., 2023) uses first-order logic with Prover9 (McCune, 2005). Verus-LM (Callewaert et al., 2025) provides a multi-paradigm framework with IDP-Z3 (Carbonnelle et al., 2022). SAT-LM (Ye et al., 2023) employs declarative prompting with SMT (De Moura & Bjørner, 2008). Logic-LM (Pan et al., 2023) supports multiple formalisms with self-refinement. Autoformalization has been studied in mathematics (Wang et al., 2018; Szegedy, 2020; Wu et al., 2022; Jiang et al., 2022). However, existing neurosymbolic systems focus on single-shot problem solving. Unlike autoformalization for mathematical statements with precise semantics, we handle policy ambiguities through a two-stage approach: the PMC resolves ambiguity with human guidance during policy creation, while the AV performs redundant translation with multiple LLMs to quantify confidence during validation. This separation enables verification of whether LLM-generated content logically follows from established policies, which is crucial for regulated industries.

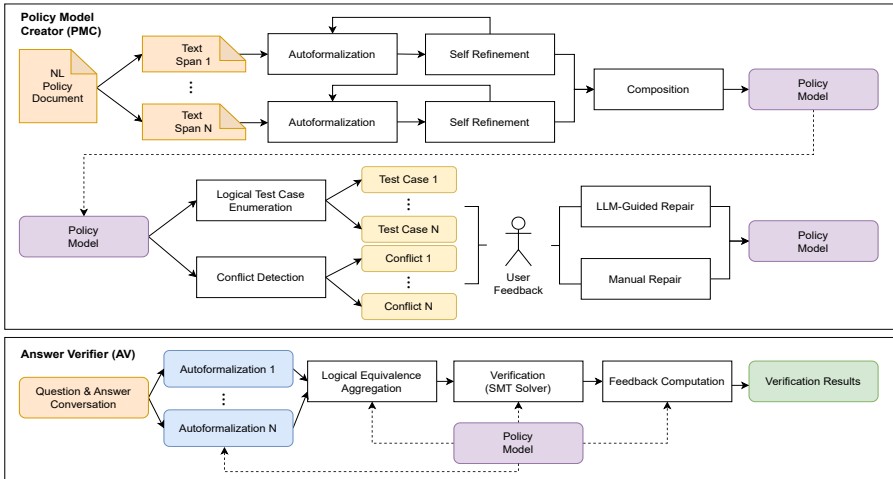

Figure 1: End-to-end architecture of LRG

## 3 METHODOLOGY

Fig. 1 shows our architecture's two main components: the PMC (§3.1) and the AV (§3.2). We illustrate our approach with an example NL policy about park admission fees:

---

**General admission**: The regular admission to the park is $50. The admission fee in the low season is 75% of the regular admission fee.

**Discount**: Seniors (age greater than 65) qualify for a 40% discount. Whenever a discount applies, there will be a $10 flat discount processing fee.

**Credit**: You can use credit for up to 50% of your final admission (1 credit for 1 dollar). However, if credits are used, then the discount rate is capped at 25%. You can purchase credit at a rate of $0.60 per credit. You can only purchase credit in increments of 5 (cost 3$).

**Tax**: A federal tax of 10% applies to the final expense.

---

Suppose a user asks "I am a senior and want to visit the park in the low season, and I have a total fund of $35.40. Can I visit the park?", and we want to verify a chatbot's answer of "No, $35.40 is not enough."

LRG tackles the verification problem in two stages. In the first stage, the PMC auto-formalizes the policy into a so-called *policy model*: a set of logic rules, expressed in SMT-LIB (Barrett et al., 2016), together with a *schema* that defines variables with their types and NL descriptions. SMT-LIB is a standardized logic language that uses prefix notation, where operators precede their arguments; e.g., "if $x$, then $y$" is written as `(=> x y)`. Fig. 2 shows snippets of the policy model. In the second stage, the AV first auto-formalizes the statement under validation into logic formulas (over the variables of the policy model), then uses an SMT solver to verify those formulas against the policy model. A snippet of the validation feedback for our example is shown in Fig. 3. The feedback includes the logic translation (with a confidence score between 0 and 1), the validation result, and further feedback explaining the result (in this case, a counter-example to validity, shown in §3.2).[1] [2]

### 3.1 POLICY MODEL CREATOR (PMC)

The PMC takes a policy document written in natural language and autoformalizes it into a policy model (§3.1.1). It also provides an array of utilities to support users in policy vetting (§3.1.2).

---

[1]The key, as indicated by the counter-example, is to use $15 worth of credit for payment: The admission fee in the low season is $37.5 (75% of $50). After applying the 25% senior discount (capped at 25% because of credit use) and adding the $10 discount processing fee, the actual admission fee becomes $38.125. This fee can be paid by combining $15 of credit (cost: $9) with a $23.125 cash payment, for an expense of $32.125. After adding a federal tax of 10%, the final expense becomes $35.3375, which is within the budget of $35.4.

[2]When we tested LLM judges (Claude Sonnet 3.7 and Opus 4.1, both with reasoning mode), they incorrectly classified this answer as valid with plausible but incorrect reasoning (see Appendix Figs 8 and 11).

| Variable | Type | Description |
|---|---|---|
| isLowSeason | Bool | Whether the designated admission day is in the low season |
| admissionFeeAfterDiscount | $\mathbb{R}$ | Admission fee after discounts are applied but before tax |

**Rule 1:** `(=> isLowSeason (= admissionFee (* 0.75 baseFee)))`

**Rule 2:** `(<= (* 2.0 customerCredits) admissionFeeAfterDiscount)`

Figure 2: Snippets of policy model (top: variable schema; bottom: rules)

### 3.1.1 AUTOFORMALIZING NATURAL LANGUAGE DOCUMENTS

To handle the size and complexity of real-world policy documents in the face of known LLM reasoning limitations around context size and distractors (Rajeev et al., 2025; Levy et al., 2024), the PMC takes a divide-and-conquer approach to autoformalize documents into logic (see Fig. 1).

The PMC first splits the input document into a set of text spans. These are processed using an incremental, refinement-guided autoformalization procedure: A language model processes each span and identifies statements that express coherent, formalizable meaning. For each statement, the LLM translates the semantic content into a list of SMT-LIB datatypes, variables, and logical constraints (*rules*). The LLM's context maintains existing declarations within a span to avoid duplicated or conflicting declarations. The complete formalization of a span is what we call a *policy unit*. If this process introduces an error (e.g., malformed syntax), we provide the invalid declarations and their failure causes to the LLM for repair in a refinement loop.

Once the PMC has formalized all text spans, it then composes the resulting policy units into a single policy model. The PMC generates textual embeddings of variables and clusters them using cosine similarity. Variables within a cluster are unified, while variables that share the same name but are not clustered are renamed. Consistent replacement of original variables with unified variables is performed for the rules of each policy unit, then the rules are aggregated, dropping syntactic duplicates. The resulting policy model is a structured representation of the document, consisting of three fields: datatypes, variables, and rules. Each variable is associated with an NL description that explains its meaning in terms of the source document, as shown in Fig. 2. This initial policy model is then vetted, as described in the next section. We measure the relationship between document size and formalized policy size in §A.1.3.

### 3.1.2 POLICY MODEL VETTING

The initial policy model that we automatically generate might contain errors, omissions, and imprecisions. Additionally, NL policies often contain ambiguities that only subject matter experts can resolve. We therefore provide users with several methods for vetting of their policy models: linting, inspection, and testing (both manual and automatic). We also develop automated repair approaches around these vetting methods.

**Linting.** We provide a linter for our policy models that checks integrity and consistency properties beyond the simple malformedness errors caught during autoformalization. First, we perform syntax-based checks. For example, we detect variables that are not used in the logical rules, and report them as warnings to the user. Second, we perform semantics checks. For example, we use an SMT solver to detect contradictory rules, which would lead to unexpected results from the AV. We show the linting report to the user as a list of errors and warnings, which can then be addressed either directly (e.g., by deleting an unused variable) or through more detailed policy inspection and repair.

**Inspection.** Manual inspection allows users to review their generated policy model and verify its correctness, similar to code review in software development. Users can examine the policy variables with their types and descriptions, as well as the logical rules themselves. We provide two views of the rules for inspection: SMT-LIB for experts and structured English for non-experts. We generate the structured English mechanically (based on templates like "if . . . then . . . ") without using an LLM to avoid potential hallucinations and imprecisions in this critical step. If users uncover an issue with a rule, they can provide NL feedback explaining what is wrong and how it should be fixed. This triggers an automatic LLM-based policy repair step that adjusts the policy model based on the provided feedback. Manual inspection provides strong correctness guarantees when all rules are carefully reviewed, but it can be challenging with large numbers of complex rules that have intricate interactions. The PMC therefore also provides testing as an additional policy vetting methodology.

---

**Algorithm 1** AV Redundant Translation

---

1: **procedure** REDUNDANTTRANSLATION($msg$, $policy$, $LLMs$)
2:   $findings \leftarrow \emptyset$; $Ts \leftarrow [Translate(msg, policy, llm)$ for every model $llm \in LLMs]$ ,
3:   **for** $T \in Ts$ **for every premise-conclusion pair** $\langle P, C \rangle \in T$ **do**
4:     $Supports \leftarrow \{T' \mid T' \in Ts$ **if** $T' \models (P \Rightarrow C)$ and $T' \not\models \neg P\}$
5:     $findings$.add($\langle P, C, conf \rangle$) where $conf = |Supports|/|Ts|$
6:   **return** $findings$

---

> **Logic Translation (Confidence: 1.0):**
> - **Premise**: `(and (= ageClass SENIOR) isLowSeason (= totalAdmissionFund 35.4))`
> - **Conclusion**: `(not isEntryAllowed)`
> **Validation Result:** *Satisfiable* (not *Valid*)
>
> **Counter-Example to Validity**:
>
> creditUnit = 3, customerCredits = 15.0, creditDollarValue = 9.0, cashAmount = 23.181, totalPayment-Available = 38.181, finalAdmissionFee = 38.125, isEntryAllowed = true, . . .

Figure 3: Snippet of validation feedback

**Testing.** Testing provides a systematic way to validate policy models through concrete examples. Similar to unit tests, test cases in the PMC are NL question-answer pairs with their expected findings (e.g., valid, invalid). Test cases can either be provided manually by users or generated automatically. The PMC offers automatic, symbolic test-case generation that leverages an SMT solver to systematically explore the state space of the policy model. Since such test cases are generated symbolically, each comes with its provably-correct expected finding. The PMC executes test cases by running the AV on them to compute the actual findings. If the actual findings do not match the expected ones, there is potentially an error in the policy model or in the AV translation. Users can then examine the policy model and translation using the information provided by the AV (e.g., the logical rules justifying the result) to root cause the issue and generate a repair.

## 3.2 ANSWER VERIFIER (AV)

The AV uses LLMs to translate natural language (typically, a question-answer pair) into premise-conclusion pairs, where premises (abbreviated as $P$) and conclusions (abbreviated as $C$) are expressed in the logical vocabulary of the policy model. For example, a statement like "Since you spent more than \$100, you are eligible for a refund" might be translated into the premise $customer\_spend > 100$ and the conclusion $eligible\_for\_refund$. Premises are the contextual facts and conditions established by the NL statement ("you spent more than \$100"). Conclusions are the logical consequences claimed to follow from those conditions ("you are eligible for a refund"). A given text fragment may be translated into multiple premise-conclusion pairs (or none), as needed to represent different claims that are asserted to follow from different conditions.

To increase translation confidence, the AV *redundantly translates* the NL statement using $k$ LLMs (Alg. 1). It compares the resulting premise-conclusion pairs semantically using an SMT solver, to estimate a confidence score for each pair. Intuitively, the confidence score of a premise-conclusion pair $\langle P, C \rangle$ is the proportion of the $k$ translations that logically entail the implication $P \Rightarrow C$.

For example, consider the text under validation in §3 and the policy model shown in Fig. 2. Suppose we run redundant translation with three LLMs that all produce identical translations containing the premise-conclusion pair shown in Fig. 3. The confidence score for this pair is $3/3$.

Now suppose one of the three LLMs produces a premise-conclusion pair with a different conclusion: `isEntryAllowed`. In this case, AV would return two distinct premise-conclusion pairs: the original pair from Fig. 3, with confidence $2/3$, and the other one with confidence $1/3$.

**Validation Feedback.** After translating the text into a list of premise-conclusion pairs with confidence scores, the AV uses the SMT solver Z3 to provide detailed, logically grounded feedback to help users understand the validation results, and (where appropriate) provide corrective guidance. We validate each translated claim (consisting of premise $P$, conclusion $C$, and confidence score) against the policy model $\mathcal{M}$, and produce one of the following findings:

- *Unknown*: The LLM is unable to translate the text into the vocabulary of the policy model.

- *TooComplex*: Either the text or the translation into SMT-LIB requires too many tokens.
- *Ambiguous*: $\langle P, C \rangle$ has a confidence score below a configurable threshold (default: 3/3).
- *Impossible* ($\mathcal{M} \vDash \neg P$): The premises alone contradict the policy model.
- *Invalid* ($\mathcal{M} \wedge P \vDash \neg C$): The conclusion must be false given the policy model and premises.
- *Satisfiable* ($\mathcal{M} \wedge P \nvDash C$ and $\mathcal{M} \wedge P \nvDash \neg C$): The conclusion is consistent with, but doesn't necessarily follow from, the policy model and the premises.
- *Valid* ($\mathcal{M} \wedge P \vDash C$): The conclusion must be true given the policy model and the premises.

For *Impossible*, *Valid* or *Invalid* findings, the findings include the relevant rules (i.e., the rules justifying the result) from the policy model, extracted from the SMT solver. For *Satisfiable* findings, the feedback returns satisfying assignments ("scenarios") demonstrating how the premises could be extended to become valid or invalid. For *Impossible*, *Invalid*, *Valid*, and *Satisfiable* findings, the findings provide sufficient information that an independent third party could use a theorem prover to re-derive the finding from the policy model. For *Ambiguous* findings, the feedback presents two differing translations along with an assignment that is satisfiable in one translation but not the other. *Unknown* findings return the relevant untranslatable text segments. Finally, logic warnings are surfaced if the premises or conclusions are always true or false irrespective of the policy rules.

Consider again the example shown in Sec. 3, where the AV is used to validate the conversation with the question "I am a senior and want to visit the park in the low season, and I have a total fund of $35.40. Can I visit the park?", and the chatbot answer "No, $35.40 is not enough." As shown in Fig. 3, the AV returns a finding of *Satisfiable* and provides a variable assignment demonstrating a concrete case in which the park admission is possible within the budget constraint (showing that the provided answer can be wrong), along with a variable assignment showing what additional information would make the answer correct.

## 4  EMPIRICAL EVALUATION

We evaluate LRG around the following research questions (RQs) to understand how effective LRG is as a guardrail and how our design choices contribute to its performance.

RQ1 (RELIABILITY OF VALIDATING LOGICAL ACCURACY):  How reliably does LRG validate logical accuracy compared to alternative baselines?

RQ2 (IMPACT OF REDUNDANT TRANSLATION):  How does redundant translation (§3.2) impact LRG's performance?

RQ3 (IMPACT OF HUMAN POLICY VETTING):  Does human policy vetting improve logical accuracy validation enough to justify the additional effort?

RQ4 (EFFECTIVENESS OF LRG'S FEEDBACK):  Is the feedback provided by LRG effective in driving improvement of LLM output?

**Metrics.**  We frame logical accuracy detection as a binary classification problem: decide whether NL statements are *Valid* or not. All metrics we report reflect this binary setting. We evaluate LRG using standard classification metrics (precision, recall, F1, accuracy), treating *Valid* as the positive class and all other categories as negative. Our primary objective, however, is to maximize recall while strictly limiting false positives across the entire pipeline. In safety-critical settings, rejecting borderline cases (*not-valid*) is still favorable: such outputs can either be refined by the answer-generating LLM using feedback from LRG, or escalated to human experts.

To capture this asymmetry, we define the *soundness* metric as the probability that content classified as valid is actually valid, computed over all decisions as $1 - \frac{\#\text{False Positives}}{\#\text{Samples}}$. High soundness ensures that incorrect content is rarely approved. When comparing alternative methods, *Valid* recall is used as a tie-breaker under the requirement of maintaining high soundness. Note that soundness is notably different from the standard performance metrics (e.g., FPR, precision, $F_\beta$) as it operates directly over the entire dataset (it is not conditional on output or label).

### 4.1  EVALUATION OF LOGICAL ACCURACY VALIDATION

**Dataset.**  We extended the ConditionalQA dataset (Sun et al., 2022) beyond its original binary classification (valid / not_answerable) into a richer evaluation set with several types of "not valid" examples. We introduce these additional categories to create variety in the dataset. The extended evaluation set includes the following categories: *Valid* (logically correct), *Invalid* (incorrect due to wrong

Table 1: Comparison of logical accuracy detection performance on CONDITIONALQA-LOGIC (Sun et al., 2022). The columns show soundness (S), false-positive rate (FPR), precision (Pr), recall (Re), F1 score (F1), accuracy (Ac), and counts of true/false positives/negatives (TP/FP/TN/FN).

| Method | S ↑ | FPR ↓ | Pr ↑ | Re ↑ | F1 ↑ | Ac ↑ | TP ↑ | FP ↓ | TN ↑ | FN ↓ |
|---|---|---|---|---|---|---|---|---|---|---|
| LRG (#3-ensemble, threshold=3/3) | **99.2** | **2.5** | 92.6 | 15.6 | 26.7 | 42.7 | 163 | **13** | **506** | 884 |
| LRG (#3-ensemble, threshold=2/3) | 98.7 | 4.0 | 91.0 | 20.3 | 33.3 | 45.4 | 213 | 21 | 498 | 834 |
| LRG (without redundant translation) | 98.6 | 4.2 | **93.8** | 31.7 | 47.4 | 52.9 | 332 | 22 | 497 | 715 |
| LLMaJ (#3-ensemble, threshold=3/3) | 98.3 | 5.0 | 92.1 | 29.0 | 44.2 | 50.9 | 304 | 26 | 493 | 743 |
| LLMaJ (#3-ensemble, threshold=2/3) | 96.3 | 11.2 | 90.1 | 50.1 | 64.4 | 63.0 | 525 | 58 | 461 | 522 |
| LLMaJ (1x Sonnet3.7) | 94.8 | 15.6 | 87.4 | 53.5 | 66.4 | 63.7 | 560 | 81 | 438 | 487 |
| LLMaJ (1x Sonnet3.7 w/ extended thinking) | 94.9 | 15.4 | 88.2 | 57.1 | 69.3 | 66.2 | 598 | 80 | 439 | 449 |
| FG Implicit span-level (Jacovi et al., 2025) | 96.4 | 11.0 | 90.5 | 52.0 | 66.0 | 64.2 | 544 | 57 | 462 | 503 |
| FG JSON (Jacovi et al., 2025) | 92.5 | 22.7 | 86.3 | 71.2 | 78.0 | 73.2 | 745 | 118 | 401 | 302 |
| FG Response-level (Jacovi et al., 2025) | 94.4 | 17.0 | 85.7 | 50.5 | 63.6 | 61.3 | 529 | 88 | 431 | 518 |
| MiniCheck (Labs, 2024) | 90.4 | 28.9 | 82.9 | 69.3 | 75.5 | 69.9 | 726 | 150 | 369 | 321 |
| RefChecker (Hu et al., 2024) | 84.4 | 47.2 | 78.9 | **87.6** | **83.0** | **76.1** | **917** | 245 | 274 | **130** |
| SelfCheckGPT (Manakul et al., 2023) | 95.0 | 15.0 | 90.5 | 71.3 | 79.7 | 75.8 | 746 | 78 | 441 | 301 |
| Logic-LM (Pan et al., 2023) | 97.4 | 7.7 | 84.0 | 20.2 | 32.6 | 44.1 | 212 | 40 | 479 | 835 |

conditions), *Satisfiable* (missing necessary conditions), *Impossible* (contradictory conditions), and *Unknown* (content that cannot be formalized, originally classified as not_answerable). These categories were created by systematically manipulating the conditional structure of original answers: removing conditions (*Valid → Satisfiable*), applying incorrect conditions (*Valid → Invalid*), or merging contradictory conditions (*Valid → Impossible*). The extended dataset (CONDITIONALQA-LOGIC) contains 349 *Valid* and 173 examples that are not *Valid* (103 *Invalid*, 52 *Satisfiable*, 4 *Impossible*, and 14 *Unknown*).

***RQ1*: Reliability of Validating Logical Accuracy.** We evaluate different validation methods on their end-to-end ability to predict validation labels for QA pairs about given NL policy documents. This evaluation tests the complete LRG pipeline (PMC plus AV) without human involvement. Table 1 reports performance comparison against alternative methods: LLM-as-Judge (LLMaJ) approaches with different prompting strategies, FACTS Grounding (FG) variants (Jacovi et al., 2025), fine-grained hallucination detection methods (Tang et al., 2024; Labs, 2024; Manakul et al., 2023; Hu et al., 2024), and the neurosymbolic Logic-LM (Pan et al., 2023).

Our evaluation shows that LRG consistently achieves the highest soundness, and the lowest false positive rate, across all methods. At the most conservative threshold (3/3), it reaches 99.2% soundness with a 2.5% false positive rate while achieving 92.6% precision. Alternative approaches did not achieve the required soundness threshold of 99%, with the second best method (in terms of soundness) being LLMaJ (#3-ensemble, threshold=3/3) at 98.3% soundness and 5.0% false positive rate, twice that of LRG. The results also highlight a clear soundness-recall tradeoff: higher soundness reduces the proportion of valid content that is accepted. For example, RefChecker has the highest recall of 87.6%, but this comes at the cost of soundness dropping to just 84.4%, the lowest of all methods. LRG's reliability comes with lower recall (15.6% for the configuration with soundness over 99%), but the tradeoff is intentional: in safety-critical domains where false approvals are far more costly than false rejections, conservatism is a necessary design choice. This makes other methods unsuitable for applications where assurance guarantees are critical. These findings underscore the importance of conservative guardrails in regulated domains, where avoiding false approvals is more valuable than maximizing coverage. The results also highlight how soundness stands apart from other metrics through its focus on minimizing false positives. For example, inspecting results of RefChecker and LRG at a low confidence threshold (1/3), we see that they exhibit higher precision, F1 score and accuracy than LRG at the maximum threshold (3/3), but at the cost of 18x and 1.5x increase in false positives, respectively. In Appendix A.1.1 we supplement this analysis with an evaluation of mitigating identified logical inaccuracies with LRG compared to the baselines.

***RQ2*: Impact of Redundant Translation.** LRG uses redundant translation (Alg. 1) to detect ambiguity and increase confidence in NL-to-logic translations. The first 3 rows in Table 1 capture the impact of redundant translation. Redundant translation improves soundness (from 98.6% to 99.2%) and reduces the false-positive rate from 4.2% to 2.5%. As discussed above, this comes at the cost of reduced recall (31.7% down to 15.6%). The tradeoff can be tuned by adjusting the confidence level "knob" in the AV: while a strict setting of 3/3 yields 99.2% soundness with only 15.6% recall, lowering the confidence to 2/3 increases recall to 20.3% at the cost of bringing soundness to 98.7%.

Table 2: Effect of human vetting on logical accuracy detection for RyanAir's customer service policy.

| Method | S ↑ | FPR ↓ | Pr ↑ | Re ↑ | F1 ↑ | Ac ↑ | TP ↑ | FP ↓ | TN ↑ | FN ↓ |
|---|---|---|---|---|---|---|---|---|---|---|
| LRG with human vetting | **100.0** | **0.0** | **100.0** | **45.5** | **62.5** | **61.3** | **20** | **0** | **18** | **24** |
| LRG without any human vetting | 96.8 | 8.7 | 84.6 | 25.0 | 38.6 | 43.6 | 11 | 2 | 16 | 33 |

## 4.2 REFINING REAL-WORLD POLICY MODELS AND ANSWERS

CONDITIONALQA-LOGIC policies are small and self-contained, and our policy autoformalization approach (§3.1.1) produces high-quality models without further review, resulting in the high soundness seen in Table 1. Real-world policy documents can be much larger and more complex than CONDITIONALQA-LOGIC policies. To understand the applicability of LRG to such real-world policies, we collected a dataset of customer-facing policy documents (ranging from refund policies to insurance policies) from six different businesses. These policies typically require human-in-the-loop refinement (§3.1.2) to capture their nuances and support sound LRG validation. We explore the impact of human vetting with a case study of one of the policies (RQ3). In a real-world setting, users may deploy LLM-based chatbots to answer questions about these policies, requiring assurance of answer correctness. To study the end-to-end impact of LRG in this setting, we evaluate iterated self-refinement of LLM answers using feedback from LRG on the dataset (RQ4).

***RQ3*: Impact of Human Policy Vetting.** We designed a case study to evaluate the support that LRG provides for policy refinement and its impact on LRG's performance as a guardrail. In our study, we create and compare two formalizations of an airline's refund policy: one using the PMC to create a policy model without additional vetting, and one that is revised using a human-in-the-loop (as described in §3.1.2). To evaluate these two formalizations, we created a test suite balanced across three sources: 1) verbatim statements from the original policy; 2) Q/A pairs generated by an LLM; 3) Q/A pairs generated by three different individuals. The expected classification labels were determined manually.

We found that clarification of ambiguities is a task that warrants human input. There were several ambiguities and edge cases in the study document, such as when a passenger passes away *on* the day of travel. The most notable ambiguity is where the policy states: "you may be entitled for a refund if your scheduled time of departure is delayed by at least 5 hours". It is unclear whether one is entitled to such a refund only if they did not travel (a prerequisite the document clarifies in other cases).

The detailed finding categories were helpful in guiding policy revisions. *Ambiguous* or *Unknown* findings typically indicated variables needed to be added or revised. *Impossible* findings, on the other hand, consistently indicated subtle rule inconsistencies and triggered rule revisions. A recurring pattern in this category is one where autoformalization fails to recognize valid exceptions to the rules that may appear in a different section of the document. For example, the document begins with a broad statement that "If your flight operated and you didn't travel, you're not entitled to a refund," but later on lists several special circumstances under which passengers may indeed qualify for a refund even if their flight operated. As shown below, LRG correctly flags the displayed question-answer pair as *Impossible* and identifies the problematic rules in the policy model.

**Question-answer pair:**
Q: My flight operated but I did not travel because I was denied boarding. Am I eligible for a refund?
A: Yes, if you were denied boarding you are eligible for a refund.

**LRG interprets the question-answer as follows:**

```
Premise: (and didFlightOperate
    (not didPassengerTravel)
    (= flightDisruptionReason DENIED_BOARDING))
Conclusion: isRefundEligible
```

**LRG judgment:** *Impossible*

**LRG returns two rules to explain this finding:**

```
1: (=> (and didFlightOperate
        (not didPassengerTravel))
    (not isRefundEligible))

2: (=> (= flightDisruptionReason
        DENIED_BOARDING) isRefundEligible)
```

The revision process was a non-trivial effort of several person-hours. As illustrated in Table 2, refinement played a central role in creating a policy that is effective as a guardrail. In fact, human vetting increased soundness to 100% and the recall to 45.5%. Note that, even though some tests were run during human vetting, about a third of the tests were held-out. Validation inaccuracies were evenly spread across all classes. Since vetted policies can be reused across future validation tasks, the cost is amortized over time, making human-in-the-loop vetting a practical and effective complement to automated formalization with human oversight.

***RQ4*: Effectiveness of LRG's Feedback.** The formally-grounded feedback that LRG provides (Section 3.2), in addition to being helpful for human vetting and policy refinement as discussed above, can also be used for automated answer revision. In Figure 4, we prompt an LLM to iteratively revise AV-judged non-*Valid* answers given LRG feedback (#3-ensemble, threshold=3/3), and plot the relative percentages of each finding type after each iteration. We can see that after just three iterations of revision, the LLM is able to go from 10.8% to 43.9% *Valid* answers. Primarily, this comes from a sharp reduction in AV-judged *Satisfiable* answers (where the answer could be true or false depending on additional context that is not stated in the answer). Given LRG's feedback, which includes a logically-derived scenario that illustrates additional context that is needed to make the answer *Valid*, the LLM is able to effectively revise these into *Valid* answers.

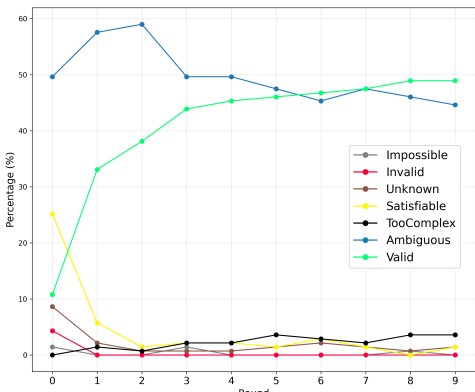

Figure 4: LRG validation finding distribution after $k$ iterations of answer revision using LRG feedback. At $k = 0$, we plot the finding distribution before any revisions.

LRG's feedback is less effective in revising *Ambiguous* and *Unknown* answers. Analysis of the revision trajectories shows that the correct revision in these cases would have been to revise the policy model, not the answer; for example, the policy model could be missing variables, leading to a failure to formalize the answer's content, or the policy model could have ambiguous variables that overlap in meaning, leading to a failure to generate a consistent formalization of the answer. As discussed in RQ3, these forms of feedback can be effectively handled by human vetting. This leads credence to the possibility of future automation of manual vetting effort by leveraging LRG's feedback to revise the policy model as well as the answer.

## 5 CONCLUSIONS, LIMITATIONS, AND FUTURE WORK

We presented LRG, a neurosymbolic guardrail that exceeds 99% (our "two nines" target) soundness when validating LLM answers against policies, an assurance unattainable by existing approaches. This soundness comes at the cost of recall, a tradeoff we believe appropriate for regulated industries. Soundness of LRG heavily depends on the quality of the policy model that it uses for validation. For this reason, LRG enables human oversight. Such oversight is particularly relevant when policies are complex or very long documents. As we have shown, when domain experts refine policy models, both soundness and recall improve significantly. Beyond metrics, our formal representations let experts resolve ambiguities in policy documents, which is something no LLM can do, as only humans with authority can provide the definitive interpretation of what was intended.

While LRG achieves strong soundness guarantees, current implementation limitations include:

- *Scalability*: Policy models from documents with hundreds of pages can include hundreds-to-thousands of rules, making them challenging for human vetting.
- *Document types*: Policies with numerical tables, cross-references, or implicit assumptions can be challenging to formalize accurately without human vetting or background knowledge.
- *Computational cost*: Redundant translation requires multiple (3) LLM calls, resulting in average 5-15 second latency and increased API cost per Q/A validation with our current implementation.
- *Autoformalization challenges*: Subtle issues like ambiguous pronouns, implicit temporal or conditional scoping, and complex nested clauses or negations can lead to incorrect formalizations that propagate through the current pipeline.
- *Human effort*: The investment for human vetting of policies, while amortized over time, remains a significant upfront cost.

Future work includes exploring automatic and confidence-aware focused vetting, fine-tuned translation models for improving accuracy and latency/costs, and improved logical formalisms to address current limitations while targeting three nines soundness and beyond. Our approach directly benefits from advances in LLMs and generative AI techniques: as models improve, their ability to formalize natural language to logic will too. We are confident LRG will inherit these improvements while maintaining the mathematical guarantees provided by symbolic reasoning.

## REPRODUCIBILITY STATEMENT

To encourage reproducibility, we included experimental details in Appendix A.2, implementation details (including detailed prompt templates) in Appendix A.3, as well as the following additional resources as supplementary material:

- The extended CONDITIONALQA-LOGIC dataset.
- Detailed artifacts for the user study from Section 4.2, including source document, policy auto-generated by PMC, policy after human vetting, and comprehensive tests.

## ETHICS STATEMENT

The authors of this work acknowledge that they have read and commit to adhering to the ICLR Code of Ethics.

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

# A APPENDIX

## A.1 ADDITIONAL EXPERIMENTS

### A.1.1 EFFECTIVENESS OF FEEDBACK FOR MITIGATING LOGICAL INACCURACIES

Table 3 sheds light on how effective is LRG's feedback compared to existing state-of-the-art methods for mitigating logical inaccuracies using the CONDITIONALQA-LOGIC dataset. We utilized a uniform experimental methodology for each method $M$: raw feedback from $M$ is incorporated into an identical prompt for answer revision through an LLM for at most 10 revision iterations. In each revision iteration $k$ ($0 \leq k \leq 10$), the answer from the previous iteration is evaluated by $M$, and if labeled as not *Valid* by $M$, a new revised answer is generated through an LLM (Claude Sonnet 3.7) by incorporating the raw evaluation feedback from $M$ in the Fig. 10 prompt. Columns 2-6 (marked under % *Valid*) report the percentage of answers classified as *Valid* by each method $M$ after different revision iterations $k \in 0, 1, 3, 5, 10$. We further evaluated the final revised answers generated after at most 10 revision iterations separately with top 3 judges from Table 1 (in terms of soundness), and report soundness and recall with respect to the corresponding judge $J$. Additionally, the CONDITIONALQA-LOGIC dataset contains 14 questions labeled as not answerable from the source text by human annotators from (Sun et al., 2021), for which we report the count of final revised answers labeled as *Valid* by method $M$ as false positives counts in the last column.

Table 3: Effectiveness of different methods in providing feedback to mitigate logical inaccuracies (as detected by the same method) on CONDITIONALQA-LOGIC (Sun et al., 2022). For each method, raw feedback is incorporated into an identical prompt for LLM-based refinement (Fig. 10). We report the percentage of responses classified as *Valid* by the method after at most 10 answer refinement iterations. Columns S & Re report soundness and recall of final revised answers evaluated separately with top 3 judges from Table 1 (w.r.t. soundness) as ground truth. Column FP$_{Human}$ reports false positives in the final revised answers for questions labeled as not answerable by human annotators.

| Method $M$ | % Valid | | | | | Judge $J$ to evaluate final revised answers after 10 revision iterations | | | | | | Not ans. |
| | | | | | | LRG (#3-ensemble) | | LLMaJ (#3-ensemble) | | FG Implicit span-level | | |
| | @0 | @1 | @3 | @5 | @10 | S | Re | S | Re | S | Re | FP$_{Human}$ |
| LRG (#3-ensemble) | 10.5 | 15.5 | 19.2 | 21.3 | 23.6 | — | | **89.9** | 33.3 | **93.9** | 72.2 | **1** |
| LLMaJ (#3-ensemble) | 23.8 | 68.2 | 73.4 | 75.9 | 77.6 | **32.2** | 30.9 | — | | 91.4 | 95.5 | 9 |
| FG Implicit span-level | 38.7 | 93.1 | 99.2 | 99.6 | **100.0** | 9.6 | **100.0** | 54.2 | **100.0** | — | | 14 |
| FG JSON | 53.6 | 93.1 | 99.2 | 99.6 | 99.6 | 11.3 | 96.6 | 46.6 | 99.2 | 78.2 | 99.5 | 14 |
| LLMaJ (1x Sonnet3.7) | 42.0 | 94.6 | 99.0 | 99.6 | 99.8 | 10.3 | 98.2 | 56.3 | 99.7 | 80.8 | **99.8** | 14 |

Examining the above results, we observe the following:

- LRG's feedback helps drive the count of *Valid* answers (as evaluated by LRG) from 55 tests initially to 123 tests after 10 iterations. LRG remains cautious and conservative when passing revised answers as *Valid*, and flags answers even in cases with minute/subtle errors or discrepancies.
- FG Implicit span-level, FG JSON, and LLMaJ (1x Sonnet3.7) methods remain much more liberal, quickly reaching to answers evaluated by them as *Valid* for $> 93\%$ tests after just 1 revision iteration, and for $> 99\%$ tests after 10 revision iterations.
- When comparing final revised answers through different judges, LRG stands out with the highest overall soundness across all top 3 judges.
- As expected, FG Implicit span-level, FG JSON, and LLMaJ (1x Sonnet3.7) methods show near-perfect recall. However, the final revised answers from these methods show major soundness gaps when evaluated with LRG or LLMaJ (#3-ensemble) judges, making them unsuitable for high-stakes tasks.
- For the 14 questions that are annotated as not answerable, LRG showed significantly lower false positives compared to other methods.
- Overall, the soundness-recall tradeoff persists across methods: LRG demonstrates the highest soundness and agreements across judges, but at the cost of lower recall rates. LRG conservative judgments and attention to minute and subtle details are well aligned for safety-critical domains where false approvals are far more costly than false rejections.

### A.1.2 UTILIZING PMC RULES BEYOND AV

In this experiment, we utilized rules generated by PMC 3.1 for CONDITIONALQA-LOGIC, and used them instead of or in addition to the source document text as in-context information for LLMaJ method. Table 4 summarizes the key results for: 1) with PMC rules instead of document text, and 2) with PMC rules in addition to the document text.

Table 4: Overall logical accuracy detection across types of in-context information for LLM baselines.

| In-Context Information | S ↑ | FPR ↓ | Pr ↑ | Re ↑ | F1 ↑ | Ac ↑ | TP ↑ | FP ↓ | TN ↑ | FN ↓ |
|---|---|---|---|---|---|---|---|---|---|---|
| LLMaJ (#3-ensemble, threshold=3/3) | 98.3 | 5.0 | 92.1 | 29.0 | 44.2 | 50.9 | 304 | 26 | 493 | 743 |
| 1) with PMC rules, without Doc | **98.9** | **3.5** | **93.5** | 24.7 | 39.1 | 48.5 | 259 | **18** | **501** | 788 |
| 2) with PMC rules, with Doc | 97.6 | 7.1 | 90.6 | **34.2** | **49.7** | **53.6** | **358** | 37 | 482 | **689** |

### A.1.3 PMC SCALING

In order to examine how PMC scales with respect to policy size, we run it over a large real-world document consisting of 274 pages of content. Each page consists of approximately 500 tokens. Figure 5 measures the number of datatypes, variables, rules with respect to document size.

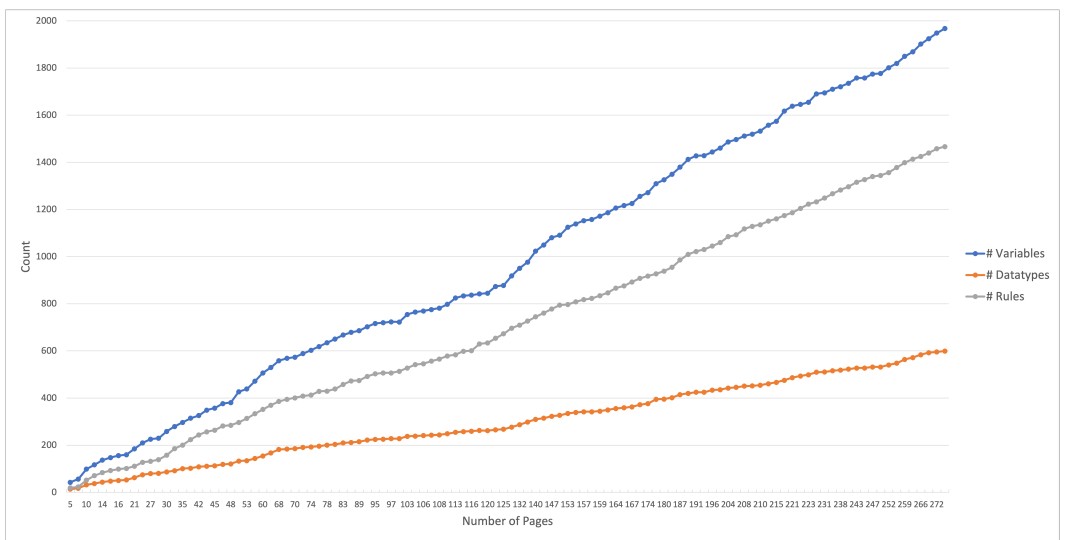

Figure 5: Number of datatypes, variables, and rules with respect to number of unique pages of text formalized, where each page is approximately 500 tokens.

We can see that PMC-produced policy size (in terms of counts of datatypes, variables, and rules) scales smoothly with document size. The overall document amounts to 600 datatypes, 1968 variables, and 1467 rules.

## A.2 EXPERIMENT DETAILS

**Dataset Details (Section 4.1).** The original ConditionalQA dataset provides binary labels (valid or not-answerable) for question-answer pairs, suited for the task of answering complex questions over long documents. However, the dataset's defining characteristic–that answers are only correct under specific stated conditions–presented a unique opportunity for more comprehensive evaluation suited for our logical accuracy detection task. We systematically leveraged this conditional structure to generate additional evaluation categories by using the relationship between answers and their associated conditions:

*Satisfiable*: Answers with non-empty conditions deliberately removed or dropped–tests whether a method can identify when necessary conditions are missing, even though the core answer content remains logically satisfiable within the document context.

*Invalid*: Answers with incorrect alternative conditions applied–evaluates a method's ability to detect when conditions directly contradict the stated answer based on the document (e.g., flipping yes/no responses while maintaining the original conditions).

*Impossible*: Contradictory yes/no answers with non-empty conditions merged–tests detection of logical impossibilities by combining mutually exclusive responses (yes/no answers) under unified condition sets.

*Valid*: Original answers with their stated conditions intact–represents the ground truth conditional answers as provided in the dataset.

*Unknown*: Questions originally marked as not-answerable in the dataset–preserves the dataset's inherent cases for which an answer cannot be given based on the source text.

This systematic augmentation transformed the original dataset into a multi-dimensional evaluation dataset for logical accuracy detection that tests conditional reasoning capabilities across various logical relationships and edge cases.

**Baseline Details (Section 4.1)**. For a fair comparison, we evaluated all methods under comparable configurations:

*LLMaJ*: For a comprehensive LLM-as-judge baseline that takes into account different validation output types comparable to LRG, we developed a customized prompt 7 with explicit instructions and details about the logical accuracy validation task. When coupled with majority voting as an ensemble of 3 (i.e., LLMaJ (#-ensemble) in Table 1), we utilized a comparable ensemble configuration as utilized in LRG (#-ensemble) (i.e., LRG with redundant translation using 3 LLM calls).

*FACTS Grounding*: We utilized the exact same prompts as presented in (Jacovi et al., 2025) using Claude Sonnet 3.7 as the LLM.

*MiniCheck*: We evaluated the method in its recommended default configuration as presented in (Labs, 2024).

*RefChecker*: We utilized the accurate context setting from (Hu et al., 2024) (input prompt provided with the document) and evaluated under joint checking of claims using Claude Sonnet 3.7.

*SelfCheckGPT*: We configured the method from (Manakul et al., 2023) with 3 samples using Claude Sonnet 3.7.

*Logic-LM*: We adapted the method from (Pan et al., 2023) and configured with Prover9[3] as the solver.

## A.3 Implementation Details

### A.3.1 Fragment of SMT-LIB utilized by LRG

LRG supports the autoformalization of natural language policy documents into quantifier-free SMTLIB with non-linear arithmetic (QF_NRIA) as shown in Fig. 6. This logical fragment allows us to express predicates over integers, real numbers, booleans, and *datatypes* (enumerated values). We restrict our approach to this logical fragment because regulatory policy documents are typically written for human consumption and thus lack complex quantification.

$$\tau := \texttt{Int} \,|\, \texttt{Real} \,|\, \texttt{Bool} \,|\, k$$

$$d := (\texttt{declare-datatype}\ k\ (v_1 \ldots v_n))\,|\,(\texttt{declare-const}\ x\ \tau)$$

$$op := +\,|\,-\,|\,/\,|\,*\,|\,=\,|\,>\,|\,<\,|\,\leq\,|\,\geq$$

$$c := integers\,|\,reals\,|\,\texttt{true}\,|\,\texttt{false}$$

$$e := x\,|\,v\,|\,c\,|\,(\texttt{and}\ e\ e)\,|\,(\texttt{or}\ e\ e)\,|\,(\texttt{not}\ e)\,|\,(\texttt{=>}\ e\ e)\,|\,(op\ e\ e)$$

$$s := (\texttt{assert}\ e)$$

$$p := d_1 \ldots d_n; s_1 \ldots s_n$$

Figure 6: The fragment of SMTLIB expressed by our formalization.

---

[3] https://www.cs.unm.edu/~mccune/prover9/

## A.4 PROMPT TEMPLATES

### A.4.1 LLM-AS-JUDGE OUTPUTS FOR RUNNING EXAMPLE

---

**System prompt**:
You are a logician that validate answer to question based on a given policy
**User prompt**:
Please consider the following policy:
<policy>
General admission: The regular admission to the park is $50. The admission fee in the low season is 75% of the regular admission fee.
Discount: Seniors (age greater than 65) qualify for 40% discounts. Whenever a discount applies, there will be a $10 flat discount processing fee.
Credit:You can use credit for up to 50% of your final admission (1 credit for 1 dollar). However, if credits are used, then the discount rate is capped at 25%. You can purchase credit at a rate of $0.60 per credit. You can only purchase credit in increments of 5 (cost 3$).
Tax: A federal tax of 10% applies to the final expense.
</policy>
Please help validate the question and answer pair as valid or not.
<question> I am a senior citizen visiting park in the low season, I have $35.4 dollars as total fund, will I be able to visit the park
</question>
<answer> No, $35.4 is not enough </answer>

---

Figure 7: LLM-as-Judge Prompt for the example in Sec. 3

---

**Valid**.

**Reasoning**:Let me calculate the cost for a senior citizen visiting the park in the low season:
Base admission price in the low season:
Regular admission: $50 low season rate: 75% of regular = $50 × 0.75 = $37.50 Senior discount: Seniors get 40% discount = $37.50 × 0.60 = $22.50 Discount processing fee: +$10 Pre-tax amount: $32.50 Tax calculation:
10% federal tax = $32.50 × 0.10 = $3.25 Final cost: $32.50 + $3.25 = $35.75 The visitor has $35.40, which is $0.35 short of the required $35.75.
Using credits wouldn't help because:
With credits, the discount would be capped at 25% (worse than the 40% senior discount) There would be additional costs to purchase the credit.
The answer is valid. With $35.40, the senior citizen does not have enough money to visit the park in the low season.

---

Figure 8: LLM-as-Judge Validation (Claude Sonnet 3.7)

**Valid**.

**Reasoning**: Looking at this problem, I need to calculate the minimum cost for a senior citizen visiting in the low season.
Let me work through the calculation:
Step 1: Low Season Admission
Regular admission: $50 low season admission: 75% × $50 = $37.50 Step 2: Senior Discount
Seniors get 40% discount Discounted admission: $37.50 × (1 - 0.40) = $37.50 × 0.60 = $22.50
Add discount processing fee: $22.50 + $10 = $32.50 Step 3: Consider Credits Using credits
would cap the discount at 25% instead of 40%, making it more expensive, so the optimal choice
is NOT to use credits.
Step 4: Apply Tax
Subtotal: $32.50 Federal tax (10%): $32.50 × 1.10 = $35.75 Final Cost: $35.75
Since the visitor has $35.40 and needs $35.75, they are short by $0.35.
Validation Result: VALID
The answer "No, $35.4 is not enough" is correct. The minimum amount needed is $35.75,
which exceeds the available funds by $0.35.

Figure 9: LLM-as-Judge Validation (Claude Opus 4.1)

### A.4.2 PROMPT TEMPLATE UTILIZED FOR LLM-BASED ANSWER REFINEMENT

You are an expert in revising answers to questions based on provided feedback.
Given a domain, a question, an original answer, and an explanation of why the original answer
is incorrect, your task is to revise the original answer based on the given explanation. Return
only the revised answer without any prefix. Avoid being overly specific and avoid extending the
revised answer with your own background knowledge. The revised answer should be consistent
with the provided source text.
DOMAIN: {domain}
SOURCE TEXT: {source_text}
QUESTION: {question}
ORIGINAL ANSWER: {original_answer}
FEEDBACK: {feedback}
REVISED ANSWER:

Figure 10: Prompt template utilized for LLM-based answer refinement.

### A.4.3 PROMPT TEMPLATE UTILIZED FOR LLM-AS-JUDGE (LLMAJ) METHOD

You are an expert document validator. Your task is to determine whether a given answer to a question is correct according to the provided policy document. When a test finishes, you're provided with a set of validation results to understand how your Automated Reasoning policy is performing. A test includes the following information:

Query and Content: A question a user might ask your GenAI application and a possible response. You define these if you manually create the test. Automated Reasoning defines these if you generated test scenarios.

Confidence threshold: The minimum confidence level for logic validation that you set for your test. This threshold determines how Automated Reasoning handles uncertainty in translating natural language to formal logic. Content that meets or exceeds the threshold is considered a high-confidence finding that can be validated with a definitive result (VALID or INVALID). Content that falls below the threshold is a low-confidence finding that's marked as TRANSLATION_AMBIGUOUS, indicating the system detected ambiguity and chose not to provide a potentially incorrect validation result.

Validation results:

Expected result: The result you expect from running the test.

Actual result: The result from running the test.

Execution result: Indicates whether the test passed. If the expected and actual results align, the test passed. If not, the test failed.

Findings: The output from an Automated Reasoning policy test is a set of findings. Findings represent factual claims contained in your test question and answer. Use these to help you understand why a test passed or failed.

Type: Translations can include a combination of claims and premises.

Premises: Provides context, assumptions, or conditions that affect how a claim should be evaluated. In question-and-answer formats, the premise is often the question itself. Answers can also contain premises that establish constraints or conditions. For example, in the question, "What numbers are divisible by 2?" and answer, "Even numbers", the premise is "numbers divisible by 2". In the statement, "When the traffic light turns green, you must go," the premises is "traffic light is green".

Claims: Factual statements that Automated Reasoning evaluates for accuracy. In a question-and-answer format, the claim is typically the answer. In a standalone statement, the claim is the fact being asserted. For example, in the question, "What numbers are divisible by 2?" and answer, "Even numbers", the claim is "even numbers".

Result: Indicates how valid a finding's claims are. For more information, see Test validation results.

Confidence: The confidence score (ranging from 0.0 to 1.0) that Automated Reasoning has in the translation from natural language to formal logic, representing how certain the system is about correctly interpreting the input text. Higher scores indicate greater certainty in the translation. For example, if a translation has a confidence of "1.0", that indicates maximum certainty that the natural language was accurately converted to formal logic. Lower confidence scores suggest the system has some uncertainty about the translation that you may want to review.

Assignments: Variable assignments from your policy that prove the finding is valid or not. Translations have logic statements that show how the natural language was converted to formal logic. These can be more complex when there is nested logic. For example, hasDogHistoryOfAggression is false.

Rules: The extracted logic from your policy that supports the finding. A test provides you with enough relevant rules from your policy to help you understand the finding result.

Figure 11: Prompt utilized for LLM-as-Judge (LLMaJ) method (Part 1/2)

The following list details possible validation results from an Automated Reasoning policy test:

VALID The claims in the model's response are logically consistent with your policy rules and can be mathematically proven correct. The response correctly follows all applicable logical constraints and the reasoning from premises to conclusions is sound.

Example: If your policy states "Employees with 1+ year of service get parental leave" and the model responds "You qualify for parental leave since you've worked here for 18 months," this would be VALID because 18 months exceeds the 1-year requirement.

INVALID The claims in the model's response contradict or violate your policy rules. The response contains statements that are mathematically provable as incorrect based on your policy's formal logic constraints.

Example: If your policy states "Employees with 1+ year of service get parental leave" and the model responds "You qualify for parental leave even though you've only worked here for 3 months," this would be INVALID because 3 months doesn't meet the 1-year requirement.

SATISFIABLE Given the information provided in the policy, whether the claims in the model's response are correct or in violation of policy rules depends on additional information that is not specified in the response. Without that additional information, the claims can neither be proven correct nor incorrect.

Example: If your policy states "Employees need 1+ year of service for parental leave AND must submit form HR-101" and the model responds "You qualify for parental leave since you've worked here for 2 years," this would be SATISFIABLE because the response correctly addresses the service requirement but doesn't mention the form requirement (without contradicting it).

IMPOSSIBLE Automated Reasoning can't make a statement about the claims. This can happen if the premises are logically incorrect, or if there is a conflict within the Automated Reasoning policy itself.

Example: If your policy contains contradictory rules like "All employees get vacation days" and "No employees get vacation days," or if the test question contains impossible premises like "What benefits do employees get if they work negative hours?", the result would be IMPOSSIBLE because the logical foundation is flawed.

TRANSLATION_AMBIGUOUS Detected an ambiguity in the translation meant it would be unsound to continue with validity checking. Additional context or follow-up questions might be needed to get translation to succeed.

Example: If your test question is "Can they take leave?" without specifying who "they" refers to, or if the model response uses ambiguous pronouns like "It depends on their situation" without clear referents, the result would be TRANSLATION_AMBIGUOUS because the system cannot reliably translate the vague language into formal logic.

TOO_COMPLEX The input contains too much information for Automated Reasoning to process within its latency limits.

Example: If your test includes an extremely long model response with hundreds of interconnected claims about employee benefits, vacation policies, health insurance, retirement plans, and performance reviews all in a single response, the result might be TOO_COMPLEX because the logical analysis would exceed processing time limits.

NO_TRANSLATIONS Identifies that some or all of the input prompt wasn't translated into logic. This can happen if the input isn't relevant to the Automated Reasoning policy, or if the policy doesn't have variables to model relevant input. If Automated Reasoning can't translate anything, you get a single NO_TRANSLATIONS finding. You might also see a NO_TRANSLATIONS (along with other findings) if some part of the validation isn't translated.

Example: If your HR policy is designed to validate employee benefits but your test question asks "What's the weather like today?" or "How do I cook pasta?", the result would be NO_TRANSLATIONS because the content is completely unrelated to your policy's domain and variables.

POLICY DOCUMENT: {document}

QUESTION: {question}

ANSWER: {answer}

Based on the document above, classify this question-answer pair into exactly one of the QA validator aggregate results

Analyze the question and answer carefully against the document. Consider: - Does the answer accurately reflect what the document states? - Are there any conditions, exceptions, or edge cases the answer fails to mention? - Is the answer always true, sometimes true, or never true according to the document?

Provide your classification as a single word from the list above in the format <answer>[...]</answer> followed by a brief explanation.

CLASSIFICATION:

Figure 12: Prompt utilized for LLM-as-Judge (LLMaJ) method (Part 2/2)