# OpenReview forum: "A Neurosymbolic Approach to Natural Language Formalization and Verification"
_ICLR.cc/2026/Conference — Submitted to ICLR 2026_

### Official Review · Reviewer_gsUz · 2025-10-27

**Soundness:** 3
**Presentation:** 3
**Contribution:** 3
**Rating:** 6
**Confidence:** 2

**Summary:**

This paper presents LOGICAL REASONING GUARDRAILS (LRG), a neurosymbolic framework addressing the stochasticity of Large Language Models (LLMs) in regulated industries (e.g., finance, healthcare) by enabling formalization and verification of natural language (NL) policies. Key contributions include: exceeding 99% soundness (near-zero false positives) on out-of-training datasets, providing auditable logical artifacts and actionable feedback for LLM output refinement, and demonstrating effectiveness on both synthetic and real-world datasets.

**Strengths:**

The two-stage neurosymbolic approach is an integration of existing techniques (LLM autoformalization, symbolic reasoning, human-in-the-loop) tailored to policy verification. Also, the authors systematically evaluate key design choices (redundant translation, human vetting) and compare against 10+ baselines, providing robust evidence for LRG’s performance.

**Weaknesses:**

1. While the soundness-recall tradeoff is justified for safety-critical domains, the 15.6% recall (at 99.2% soundness) is notably low. The paper could explore mitigation strategies to reduce false negatives, or hybrid filtering to prioritize high-impact valid cases—without compromising core soundness.
2. The paper notes that human vetting amortizes over time but provides limited quantitative data on the upfront cost (e.g., hours per policy, number of experts required, error reduction per vetting iteration).
3. The paper compares LRG to neurosymbolic methods (e.g., Logic-LM) but could more clearly articulate how its two-stage design and policy-specific focus differ from general-purpose logical reasoning frameworks.

**Questions:**

First of all, I am not very specific in this domain. Here are some of my superficial questions. Of course, I will also refer to other reviewers for their scores.

1. For the low recall (15.6% at 99.2% soundness), have you explored adaptive thresholding or combining LRG with lightweight pre-filters to prioritize likely valid cases?
2. For large policy models, do you have strategies to streamline human vetting?
3. The feedback mechanism struggles with Ambiguous/Unknown answers due to policy model gaps. How to solve it?
4. How does LRG perform on policies with implicit temporal/conditional scoping (mentioned as an autoformalization challenge)?
5. In the real-world case study, did you evaluate LRG’s impact on chatbot deployment?

---

### Official Review · Reviewer_4hVN · 2025-10-30

**Soundness:** 2
**Presentation:** 3
**Contribution:** 2
**Rating:** 2
**Confidence:** 3

**Summary:**

This paper introduces Logical Reasoning Guardrails (LRG), a neurosymbolic framework combining LLMs and symbolic reasoning for formalizing and verifying whether natural language outputs comply with institutional policies expressed in natural language. It proposes a two-stage approach: (1) the Policy Model Creator (PMC), which auto-formalizes documents into formal policy models vetted with optional human oversight, and (2) the Answer Verifier (AV), which uses LLMs to formalize validation statements and an SMT solver to provide logically-grounded verification. Empirical evaluation on extended ConditionalQA-benchmark variants and real-world policies demonstrates soundness above 99%, outperforming neural and neurosymbolic baselines, especially in minimizing false positive rates.

**Strengths:**

- The LRG system is clearly presented, with a natural motivation for regulated/high-assurance domains where reliability and logical soundness are paramount
- The framework offers a bi-directional benefit. By attempting to formalize natural language policies, the system can surface ambiguities and inconsistencies within the original source text (as noted in Lines 409-417), effectively using the formalization process itself as a tool for improving the clarity of the natural language policies. This is quite interesting.

**Weaknesses:**

- Feasibility of Formalizing General Natural Language: The paper's central premise rests on the ability to formalize general natural language, but it does not sufficiently discuss the fundamental limitations of this endeavor. While formalization is well-understood for constrained domains like mathematics, its applicability to open-ended natural language is not guaranteed and can face significant, even philosophical, challenges regarding expressiveness and interpretation. The core assumption that a wide range of institutional policies and LLM outputs can be faithfully formalized is not robustly defended or explored.

- Correctness of Formalization: Related to the first point, the evaluation methodology does not directly measure the correctness of the formalization itself. The experiments report on the final classification accuracy of the verification task, which is an end-to-end metric. This setup treats the formalization process as a black box. Without a direct evaluation of how accurately the Policy Model Creator translates natural language into formal logic, it is difficult to ascertain the validity of this crucial intermediate step.

- Scalability and Human Effort Requirements: As acknowledged in Section 5, human vetting remains a practical bottleneck for scaling to long, complex policies with hundreds or thousands of rules. Even with automation, the upfront human review cost is significant. This means the system is not “push-button” for enterprise adoption, which challenges claims of broad applicability. The need for experts to resolve ambiguous or conflicting formalizations is a serious limitation that is not adequately addressed by the current implementation.

- Empirical Evaluation Limitations: While the policy model creation is tested on the ConditionalQA-LOGIC dataset, these are largely concise, synthetic examples. The experiments on genuinely real-world, deeply nested, or cross-referenced policies receive significantly less attention, and the reported real-world policies study is limited in scale and complexity. The paper does not rigorously quantify the performance gap or challenges when moving from smaller synthetic policies to larger, more complex ones.

- Recall-Soundness Tradeoff and Practicality: As shown in Table 1, achieving the highlighted 99.2% soundness comes at a steep cost: recall plummets to 15.6%. While the paper correctly observes that this may be appropriate for highly regulated domains, it substantially limits practical adoption for broader document understanding tasks where coverage is also necessary.

**Questions:**

1. Could the authors elaborate on the theoretical or empirical basis for assuming that formal methods can be effectively applied to a broad spectrum of general natural language policies and outputs?  What are the perceived boundaries or limitations of expressiveness for your formalization approach?

2. How could one design an experiment to directly quantify the correctness of the intermediate formalization step (i.e., the mapping from natural language to the logical model), separate from the final verification task accuracy? This seems critical for validating the core contribution.

3. Considering the acknowledged bottleneck of human vetting and the limited scale of the real-world experiments, what is the envisioned path to demonstrate that LRG can scale effectively to complex, enterprise-level policy documents?

---

### Official Review · Reviewer_N5bu · 2025-10-31

**Soundness:** 2
**Presentation:** 3
**Contribution:** 3
**Rating:** 2
**Confidence:** 5

**Summary:**

This paper presents Logical Reasoning Guardrails (LRG), a two-stage neurosymbolic framework for verifying LLM-generated content against formal policy models. The Policy Model Creator (PMC) translates natural language policies into SMT-LIB logic, while the Answer Verifier (AV) validates claims using redundant translation and SMT solving. The authors claim to achieve 99.2% soundness, exceeding existing approaches.

While the problem is important and the technical approach is sound, the evaluation is severely limited in scope, missing important LLM+solver baselines and standard benchmarks. The paper evaluates primarily on a single custom dataset, omits comparison with directly competing neurosymbolic methods, and provides insufficient evidence that the 99.2% soundness (at 15.6% recall) justifies the computational cost and human effort required.

**Strengths:**

1. Verifiable guardrails for regulated industries are a timely and practically relevant challenge.
2. The combination of autoformalization, redundant translation, and SMT verification is well-motivated and well-implemented.
3. 99.2% soundness (2.5% FPR) represents strong performance on the primary metric, important for safety-critical applications.
4. Human-in-the-loop vetting, multiple finding categories, and actionable feedback demonstrate thoughtful system architecture.
5. The system produces auditable logical explanations and counter-examples, valuable for debugging and trust.
6. I liked the honest limitations discussion. Authors acknowledge scalability, cost, and human effort challenges - often missing from LLM+Solver papers.

**Weaknesses:**

In its current form, the paper does not meet the bar to be published as a full paper at ICLR. Please consider the following as constructive feedback on how to improve it during the rebuttal process. I will increase my ratings if you can conduct all these experiments.

1. **Limited Evaluation Scope**: The paper evaluates primarily on ONE custom modified dataset (ConditionalQA-Logic), with only ONE real-world case study (44 test cases).
   1. The paper is missing evaluation on LogicNLI (Tian et al), StrategyQA (Geva et al.), ProntoQA (Saparov et al.), FOLIO (Han et al.), ProofWriter (Tafjord et al.), EntailmentBank (Dalvi et al.), BIG-Bench, or any established logical reasoning benchmarks for LLM+Solver papers. (**Please evaluate on at least 3 aforementioned datasets**)
   2. The proposed evaluation on the ConditionalQA-Logic uses systematically manipulated errors (deliberately removing conditions, flipping yes/no) rather than natural LLM mistakes. While this could be justified as a separate contribution, it cannot be the primary result. The authors should benchmark on standard datasets used in the literature mentioned above.
   3. The authors discuss six real-world policies collected but evaluate only one. Perhaps these could be explored qualitatively in the appendix (Weakness 3)
   4. Table 2 numbers are not statistically significant. 44 examples are insufficient to claim 100% soundness. Please avoid claims that are not backed up rigorously.

2. **Missing Baseline Comparisons**: Please compare your method via benchmarking against (**benchmarking against a subset is acceptable, but compare your methodology against all to show why you are different/better**) the mentioned papers and other well-known techniques in the LLM+Solver paradigm, namely: (1) LINC (Olausson et al. Cited, not compared.), (2) Proof of Thought library (very similar yet not cited or compared, Ganguly et al. does both JSON and SMT program generation), (3) SAT-LM (Ye et al. cited, not compared), and (4) Verus-LM (Callewaert et al. cited, not compared). This is important because LogicLM is compared in Table 1, where it shows soundness & FPR numbers outperforming 9/10 baselines. I am looking for better answers to the following related questions:
   1. Without the aforementioned analysis, the claim in Line 58 - "Exceeds 99% soundness... unattainable by existing approaches". This is not backed up with sufficient evidence.
   2. Line 61 claims the method "outperforms" on soundness but achieves only 1/3 to 1/6 the recall of alternatives. For example, in Table 1, RefChecker achieves 87.6% recall versus LRG's 15.6% recall. Moreover, at 99.2% soundness, the system achieves only 15.6% recall, rejecting 84.4% of valid content. While this is acknowledged as a trade-off, no analysis shows how this recall can be improved by examining threshold-agnostic measures such as the area under the risk-coverage curve.
   3. Soundness, with its particular definition as a metric, is proposed for the first time in this paper. Please consider emphasizing on other well-accepted metrics such as False Positive Rate.
3. **Need for Domain-Specific Analysis**: The paper opens with applications to legal, medical, and financial compliance. Qualitative case studies from two of these domains would significantly strengthen the paper. Please also discuss qualitatively, through multiple autoformalizations using PMC, the inherent uncertainty in outputs of LLM-driven auto formalizations (Ganguly et al. 2025) and how that may impact the correctness of your pipeline in the absence of human review.

**Questions:**

Please see weaknesses above.

---

### Official Review · Reviewer_7Twk · 2025-11-02

**Soundness:** 2
**Presentation:** 2
**Contribution:** 2
**Rating:** 2
**Confidence:** 4

**Summary:**

This paper introduces LRG (Logical Reasoning Guardrails) — a neurosymbolic framework designed to ensure high-assurance logical accuracy when LLM-generated answers rely on natural-language policy documents. The approach separates: Policy Model Creator (PMC) , Offline autoformalization of NL policies into SMT-LIB logic, with optional human vetting (linting, rule inspection, symbolic test generation).
Answer Verifier (AV): Online validation that translates QA content into logical premise-conclusion pairs using multiple LLMs for redundancy and uses an SMT solver to deliver explainable verification based on sound logical inference.

**Strengths:**

- Provides logical justification / counterexamples for every verdict
- Improves LLM responses by iteratively refining answers
- Standardized logical formalism with SMT-LIB

**Weaknesses:**

- Recall is low: A Conservative guardrail can block many correct responses
- Human vetting is time-intensive for long/complex policies
- Formalization errors in PMC propagate into AV despite validations
- Numerical tables, implicit temporal constraints, and nested logic are still hard for autoformalization

My primary concern is that the core technical contributions seem closer to a well-engineered pipeline than a novel methodology advance. The system components, like natural language autoformalization, symbolic consistency checking, redundant translation, and SMT-based verification, are mostly variations of existing neurosymbolic techniques. Much of the contribution lies in integration and workflow design, particularly the human-in-the-loop vetting tools (linting, structured English rules, symbolic test generation), which, although useful, feel incremental from a research innovation standpoint rather than proposing new theoretical insights or algorithmic techniques.

Additionally, the writing could benefit from a more polished presentation and better articulation of novelty. Certain implementation decisions appear heuristic without rigorous justification or ablation. It would help to more clearly highlight what is conceptually new versus what is assembling existing modules. Finally, while the focus on soundness over recall is reasonable for high-assurance domains, the very low recall raises questions about real usability and practical tradeoffs.

**Questions:**

- How does performance scale with very large enterprise policies (hundreds of pages)?
- Can policy vetting be partially automated, e.g., with rule clustering or active-learning review?
- Could reinforcement learning or retrieval improve recall while preserving soundness?
- What guarantees ensure that the autoformalized policy faithfully captures human legal intent?
- How sensitive is the system to domain terminology shifts (e.g., health vs. finance policies)?

---

### Official Review · Reviewer_nyXc · 2025-11-11

**Soundness:** 3
**Presentation:** 2
**Contribution:** 2
**Rating:** 4
**Confidence:** 5

**Summary:**

This paper presents LRG (Logical Reasoning Guardrails), a two-stage neurosymbolic framework that aims to bring formal logical verification into the workflow of LLMs applied to compliance, legal, and policy-related question answering.

The framework consists of:
(1) Policy Model Creator (PMC):
A module that automatically converts long natural-language policy documents into formal logical models (SMT-LIB rules), while supporting human-in-the-loop review, linting, testing, and auto-repair.
(2) Answer Verifier (AV):
A runtime component that translates QA instances into multiple candidate logical assertions using diverse LLMs, checks their semantic equivalence, and verifies each using an SMT solver (Z3). It outputs structured verdicts such as Valid, Invalid, Satisfiable, Impossible, Ambiguous, or Unknown, accompanied by explicit reasoning evidence and counterexamples.
On the ConditionalQA-Logic benchmark, LRG achieves 99.2% soundness (false-positive rate 2.5%) under a strict 3/3 consistency threshold, significantly outperforming strong baselines. A real-world case study on airline refund policies further demonstrates that human-assisted policy repair improves soundness to 100%.
Overall, LRG provides a practical, auditable pipeline for enforcing logical correctness and interpretability in LLM-based systems.

**Strengths:**

- The proposed LRG framework effectively integrates a human-in-the-loop refinement process in the Policy Model Creator (PMC), enhancing the accuracy and interpretability of autoformalized policy models, which I think should be an essential step for ensuring the soundness of future verification.

- The Answer Verifier (AV) introduces redundant translation and semantic voting across multiple LLMs to improve both syntactic and semantic reliability during natural language-to-logic formalization.

- Unlike binary verification schemes, LRG supports multiple feedback categories (Valid, Invalid, Satisfiable, Impossible, Ambiguous, etc.), providing users with explainable, logically grounded feedback and counterexamples that facilitate iterative refinement.

- The work is supported by extensive experiments across diverse neurosymbolic baselines. The analysis of soundness, recall, and redundancy impact provides a well-rounded assessment.

**Weaknesses:**

- Despite achieving 99.2% soundness, LRG’s recall remains low (15.6%). The experiments primarily involve moderately complex domains (e.g., park admission, airline refund). Evaluating LRG on domains with nested logical dependencies—such as healthcare or financial compliance—would better demonstrate its scalability and generality. Nonetheless, this limitation may represent a natural boundary of current verification paradigms and could be left for future work.

- The comparison with prior methods (e.g., methods in Table 1) focuses primarily on soundness and recall. Further analysis in terms of latency, token cost, and scalability is also necessary (in my view).

- It also remains unclear whether the main novelty lies solely in the fine-grained feedback mechanism and human-in-the-loop formalization, compared with other methods given in the paper, like Logic-LM (as Logic-LM makes formalization fully automated and only reports the deterministic results). The authors should make the discussion about the difference between existing works deeper.  Currently, it is not so clear in the paper. The paper also lacks delineated conceptual distinctions between LRG and other emerging verification approaches, such as RvLLM (which also provides some policy model for verification), proposed earlier this year.

- While the system introduces multiple fine-grained feedback types, concrete examples are sparse. Providing at least one example for each feedback category would help readers understand their practical role in guiding refinement.

- Since this work includes human-in-the-loop for the policy model generation, providing a user study would make the paper more solid.

Other Minor Issues
- The expression "Satisfiable ($M\wedge P\nvDash C$ and $M\wedge P\nvDash\neg C$)" is confusing — it seems that "and" should be replaced with "or"? If this interpretation is correct, would a simpler and clearer notation, such as "$M\wedge P$", be sufficient?
- Figures 2 and 3 use inconsistent variable or predicate names; aligning them under a single, coherent example would enhance readability and consistency.

I may revise my final score depending on the authors’ response.

**Questions:**

See the weaknesses above.

---

### Author Response · Authors · 2025-12-01

We sincerely thank all reviewers (nyXc, 7Twk, N5bu, 4hVN, and gsUz) for your thorough, thoughtful, and constructive feedback. We appreciate the time and effort you invested in evaluating our work, and we recognize that many of your criticisms are valid and provide valuable direction for improving this research.

Your reviews identified several common themes, including evaluation scope, baseline comparisons, and the recall-soundness tradeoff. We’ve compiled responses to specific issues below:

**On the recall-soundness tradeoff**

All reviewers brought up the issue of low recall in our work, which we acknowledge as an opportunity for improvement. We have made an intentional design choice to prioritize soundness, for safety-critical use cases.  Having said that, as shown in our case-study, recall can improve substantially with a carefully designed policy (where we achieve 45% recall with human policy refinement).

**On comparisons to additional existing works**

We thank the reviewers to pointing us to some relevant work for possible comparison (LINC, SAT-LM, Verus-LM , Proof of Thought, RvLLM). We tried several of these works for our experiments, however we found each of them requires non-trivial adaptations to enable fair comparisons (they cannot be used in our setting out of the box). We compared to LogicLM as a representative example of this class of work.

**No existing benchmark matches our problem**

To the best of our knowledge, there are no external benchmarking datasets that exactly match the problem we are solving: verification of questions and answers against policy documents. This is a new problem formulation. The cited benchmarks (LogicNLI, StrategyQA, FOLIO, ProofWriter, etc.) focus on general logical reasoning or knowledge retrieval, not policy-grounded compliance verification.

**Why ConditionalQA-Logic?**

We selected ConditionalQA as our base dataset because it focuses on conditional reasoning ("if X then Y"), which is fundamental to policy compliance. Policies are inherently conditional: "If customer cancels within 24 hours, then full refund applies." Our modifications transformed it from abstract logical reasoning into policy-grounded scenarios, making it directly relevant to our compliance verification use case.

**Why did we extend ConditionalQA with systematic errors?**

Our choice to introduce systematic/synthetic errors into the benchmark rather than to use a LLM to produce more natural mistakes provides controlled evaluation of specific failure modes. Systematically removing conditions tests whether the system catches incomplete policy application, while flipping yes/no tests whether the system detects logical contradictions. This gives us diagnostic precision. We know exactly what type of error we're testing for at each step.

This systematic approach provides coverage guarantees that LLM-generated errors cannot. LLM-generated errors are unpredictable and may miss critical failure types that are rare but important. Systematic errors ensure we test all important logical error categories comprehensively. Additionally, synthetic errors are deterministic and reproducible, allowing fair comparison across methods and providing clear ground truth for what should be caught.

**Why not real-world production data?**

We cannot publish real-world examples our system encounters in production (due to confidentiality). Our customer case study provides a natural setting with real-world complexity, while ConditionalQA-Logic provides a quantitative assessment.

**Statistical significance of 44 examples**

We acknowledge the airline case study with 44 examples is limited for strong statistical claims. We should tone down the 100% soundness claim for this example, or provide confidence intervals.

**What about FOLIO specifically?**

FOLIO uses first-order logic (FOL) but our approach currently only supports a quantifier-free subset of FOL. As such it is not able to represent some of the FOLIO problems. In the future we are considering extending our approach to unrestricted FOL.

---

### Meta-Review · Area_Chair_DUsL · 2026-01-12

**Summary:**

The reviewers raised many important concerns and unfortunately the authors were unable to respond to the concerns before the discussion freeze. The primary concerns that justify a rejection are, poor recall and limited empirical evaluation. There is consensus among the reviewers that the paper does not provide sufficient experiments to justify the additional computation costs of verification compared to existing approaches which do not use verification. The authors provided a global response that does not sufficiently address the concerns of the reviewers. For example, on comparisons with existing works the authors mention that previous work requires non-trivial adaptations to enable fair comparisons, but do not provide any additional information about what kind of adaptions would be required, or justify how complicated this would be. Overall, the paper does not meet the bar for publication.

**Reviewer Concerns:**

The rebuttal did not sufficiently any of the reviewer concerns.
Outstanding concerns: poor recall and limited empirical evaluation.

**Reviewer Scores:**

None of the reviewers are likely to have changed their scores.

---

### Decision · Program_Chairs · 2026-01-26

Reject